# MPN: The Molecular Drivers of Disease Initiation, Progression and Transformation and their Effect on Treatment

**DOI:** 10.3390/cells9081901

**Published:** 2020-08-14

**Authors:** Julian Grabek, Jasmin Straube, Megan Bywater, Steven W. Lane

**Affiliations:** 1Cancer Program, QIMR Berghofer Medical Research Institute, Brisbane, QLD 4006, Australia; julian.grabek@QIMRBerghofer.edu.au (J.G.); Jasmin.Straube@qimrberghofer.edu.au (J.S.); Megan.Bywater@QIMRBerghofer.edu.au (M.B.); 2Faculty of Medicine, The University of Queensland, Brisbane, QLD 4072, Australia; 3Cancer Care Services, Royal Brisbane and Women’s Hospital, Brisbane, QLD 4029, Australia

**Keywords:** MPN, myeloproliferation, JAK2, CALR, MPL, driver mutations, leukemic transformation, IFNα, chromatin modifiers, spliceosome, DNA methylation, tumour suppressors, transcriptional regulators

## Abstract

Myeloproliferative neoplasms (MPNs) constitute a group of disorders identified by an overproduction of cells derived from myeloid lineage. The majority of MPNs have an identifiable driver mutation responsible for cytokine-independent proliferative signalling. The acquisition of coexisting mutations in chromatin modifiers, spliceosome complex components, DNA methylation modifiers, tumour suppressors and transcriptional regulators have been identified as major pathways for disease progression and leukemic transformation. They also confer different sensitivities to therapeutic options. This review will explore the molecular basis of MPN pathogenesis and specifically examine the impact of coexisting mutations on disease biology and therapeutic options.

## 1. Introduction

Myeloproliferative Neoplasms (MPN) are clonal haematological disorders that lead to overproduction of cells derived from the myeloid lineage. Specific driver mutations within stem cells and myeloid progenitors provide cytokine-independent or -hypersensitive proliferative signals leading to the overproduction of myeloid cells. Here, we categorise the genetic changes that lead to these disorders and that contribute to progression of disease.

The 2016 WHO classification of MPN includes *BCR-ABL1*-positive chronic myeloid leukaemia (CML), polycythemia vera (PV), essential thrombocythemia (ET), primary myelofibrosis (PMF), chronic neutrophilic leukaemia, chronic eosinophilic leukaemia and MPN-unclassifiable. For the purpose of this review, we will focus on the more common MPNs of PV, ET and PMF as they share similar driver mutations. CML is a biologically distinct entity and will not be considered in this review.

PV is characterized by elevated red cell production including increased haemoglobin concentration, increased haematocrit and/or increased red cell mass. The vast majority carry a Janus Kinase 2 *JAK2^V617F^* mutation with a rare subset having an JAK2 exon 12 mutation.

ET is characterized by sustained thrombocytosis and increased megakaryocyte numbers. The most common driver mutation is *JAK2^V617F^* in 50–60% of cases, with Calreticulin (CALR) mutations in 30% and MPL (Myeloproliferative Leukemia) mutations at <5% of cases.

PMF is characterized by proliferation of abnormal megakaryocytes and granulocytes with deposition of fibrotic tissue in the bone marrow and extramedullary haematopoiesis. The driver mutations most commonly identified are *JAK2^V617F^* in 50–60%, with *CALR* in 30% and *MPL* in 8% of cases [1].

MPNs can have a variable disease courses, with PV and ET having a more indolent course; however, the progression can be variable with the overall survival for PV ranging from 13.5 to 24 years and ET ranging from 11 to 22.6 years. Progression to myelofibrosis (MF) over a period of 20 years occurs in 26% of PV and 19.9% of ET. The incidence of leukemic transformation over a period of 20 years is 7.9–17% for PV and 8.1% for ET [2,3].

The purpose of this review is to examine the prognostic scoring systems utilized in MPNs. We will explore the driver mutations that are diagnostic hallmarks of MPN, their pathogenesis behind disease and their phenotypic variation accounting for the same mutations in different disease phenotypes. We will then review coexisting mutations that corroborate driver mutations leading to progression. Finally, we explore the therapeutic options in MPN and how they impact the pathogenesis of MPN.

## 2. Prognostic Scoring Systems

Several prognostic scoring systems have been implemented to predict outcome in MPN with specific focus on each disease phenotype. PV and ET scoring systems focus on thrombotic risk and overall survival, whereas MF scoring systems focus on the risk of leukemic transformation and survival. 

For PV, a two-tiered stratification system using age and previous thrombosis to group patients into high or low risk for thrombotic complications guides risk adapted therapy. Overall survival scores initially incorporated both age ≥ 60 years and previous thrombosis. An International Working Group (IWG) included leukocytosis (≥15 × 10^9^/L) to group patients into low, intermediate or high risk [4]. More recent data showing the prognostic impact of *JAK2^V617F^* allele burden [5]; cardiovascular risk [6]; and high risk mutations of *ASXL1*, *SRSF2* and *IDH2* [7] are not yet incorporated into a prognostic scoring system routinely used in PV [8]. The recently proposed Mutation enhanced International Prognostic Scoring System (MIPSS-PV) adds adverse mutations (specifically *SRSF2*), further enhancing prognostication (Table 1) [9].

The International Prognostic Score for ET (IPSET) is subdivided into factors that influence thrombosis risk and survival. Risk factors for thrombosis include age ≥ 60 years, previous thrombosis and *JAK2^V617F^* mutation, whereas survival incorporates age ≥ 60 years, leukocytosis (≥11 × 10^9^/L) and previous thrombosis [10]. Recent data shows high-risk mutations in ET of *SH2B3*, *SF3B1, U2AF1, TP53, IDH2* and *EZH2* and show an effect independent of the current scoring systems [7]. Thus, the MIPSS-ET has been recently developed to include these updated criteria (Table 1) [9].

Several prognostic scoring systems have been developed for MF. Initially, the International Prognostic Scoring System (IPSS) was developed in 2009, followed by the dynamic IPSS (DIPSS), with the advantage of the DIPSS being able to be performed at any time point and with the IPSS being only validated at diagnosis [11]. The advent of increased molecular prognostic markers has led to the development of scoring systems that incorporate these such as the MIPSS70 (mutation enhanced IPSS) or rely solely on cytogenetic and molecular markers, i.e., GIPSS (genetically inspired prognostic score system). The most recent iteration of the MIPSS70+ version 2.0 is detailed in Table 2 together with the DIPSS (Table 3) that is used in the majority of clinical trials in MF [12,13]. 

Due to differences in disease biology and phenotype between PMF and post PV/ET MF [14], a separate scoring system is utilized, the MYSEC-PM, which includes age at diagnosis of secondary MF, Hb < 110g/L, platelets < 150 × 10^9^/L, unmutated *CALR*, circulating blasts of 3% and constitutional symptoms [15]. The role of secondary molecular lesions in post PV/ET MF is still under evaluation. 

Movement towards a personalized classification and prognosis system has further been enhanced by the work of Grinfeld et al., identifying distinct genetic subgroups and correlating them with disease phenotype and patient outcome. The personalized calculator at https://www.sanger.ac.uk/science/tools/progmod/progmod/ incorporates the results of this work to predict a patient’s progress at dynamic timepoints presenting survival metrics and information on disease biology (chronic phase, myelofibrosis and acute myeloid leukaemia (AML)) [16].

## 3. Driver Mutations

Mutations in *JAK2*, *CALR* and *MPL* are mutually exclusive [17]. Each driver mutation has unique disease kinetics that lead to clone expansion and acquisition of coexisting mutations. The frequency of each driver mutation is displayed in Figure 1. 

## 4. JAK2 (Janus Kinase 2)

The Janus Kinase 2 gene is located on chromosome 9 at locus p24.1 [18]. The gene consists of a C-terminal JH1 (Jak Homology) domain that is highly conserved region containing activation loop, primary phosphorylation sites and ATP-binding site. The JH2 pseudokinase domain is structurally similar to JH1 but lacks any catalytic activity. The SH2 (Src Homology 2)-like domain and adjacent FERM (4.1/ezrin/radixin/moesin) domain for the N-terminal facilitates JAK2 interaction with cell surface receptors [19]. *JAK2* has two main interactions with activation loop of JH1 undergoing phosphorylation and release of the JH2-mediated inhibition of kinase activity to facilitate *JAK2*-mediated signalling [20]. 

Somatic mutation of *JAK2* with a G to T base change at nucleotide 1849 in exon 14 of the pseudokinase domain JH2 of the gene results in an amino acid substitution of valine to phenylalanine at codon 617 (*JAK2^V617F^*) [21,22,23,24]. Mutation within the pseudokinase domain prevents its normal auto inhibitory function and leads to constitutive kinase activation [25]. Constitutive activation leads to cytokine hypersensitivity and cytokine receptor activation independent of ligand binding [26]. *JAK2* phosphorylation leads to downstream activation of signalling through STAT, MAPK, PI3K and Akt pathways [25] with STAT signalling in particular showing increased activation of proliferation genes and pro-survival genes such as Bcl-XL [27].

Whilst *JAK2* mutation is associated with myeloproliferative neoplasms, it has been identified in asymptomatic clonal haematopoiesis (CH), a, expansion of clonal haematopoietic cells but without diagnostic criteria for a haematologic disease present [28]. The progression of clonal haematopoiesis through to the development of disease is principally determined by *JAK2^V617F^* mutant allele burden [29]. Knock in mouse models using a small mutant *JAK2^V617F^* population showed progressive expansion of the *JAK2^V617F^* clone with expansion of multipotent progenitors (MPP) and megakaryocyte progenitors (MkP) driving the development of disease phenotype [30]. The allele burden has also been implicated in determining disease phenotype with lower allele burden seen in essential thrombocythemia compared to polycythemia vera [31]. The mutation often occurs initially as heterozygous with homozygous mutations associated with progression with loss of 9p during mitotic recombination leading to copy neutral loss of heterozygosity [32]. A rise in allele burden has been correlated with progression to myelofibrosis in mouse models [33] and is consistent with observations of higher allele burden in MF patients compared to PV and ET [31]. Homozygous mutations show erythropoietin (EPO) and thrombopoietin (TPO)-independent growth in cell line experiments, whilst heterozygous mutations show only TPO independence [34] consistent with a higher frequency of homozygous mutations seen in PV. The frequency of homozygous mutations varies by 25–30% in PV patients and 2–4% in ET; homozygous mutations are associated with a greater risk of progression to myelofibrosis [35]. Homozygous subclones in PV are distinct from ET as the homozygous clone often expands early in the disease to become the dominant clone, whereas in ET, homozygous clones are rarely the dominant clone [36]. The heterozygous clones in ET and PV exhibit differential interferon (IFN) signalling and STAT1 phosphorylation. Increased STAT1 activity in CD34 progenitors produce an ET-like phenotype, whereas downregulation leads to a PV-like phenotype [37]. STAT1 knockout mice showed increased BFU-E and reduced colony forming megakaryocyte precursors favouring a PV phenotype. In contrast, STAT1 activation favours higher levels of IFNgamma constraining erythroid differentiation and increasing megakaryocyte proliferation [38]. The balance between STAT1, STAT3 and STAT5 accounts for some of the phenotypic differences in *JAK2^V617F^* positive MPN [39]. In *JAK2^V617^* positive mice, Stat3 deletion in haematopoietic cells enhances thrombocytosis, accelerates fibrosis and decreased overall survival [40]. STAT5 is critical to the pathogenesis of *JAK2^V617F^*-positive PV with Stat5 conditional knockout mice showing normalization of counts and abrogation of erythropoietin independence in erythroid colonies [41].

Myelofibrosis is distinct from PV and ET as it represents progression to a pro-inflammatory and profibrotic microenvironment beyond expansion of the MPN clone and increasing allele burden. STAT3 is critical to this process with production from both the MPN clone and microenvironment promoting MF with only pan-haematopoietic deletion of STAT3 abrogating the phenotype [42]. Cytokines levels are higher in myelofibrosis compared to PV and ET, with higher IL-2, sIL-2Rα, IL-6 and tumour necrosis factor α (TNFα) production. IL-2 in particular was also higher in ET, where a greater degree of fibrosis was present [43]. Decreased GATA1 expression leads to impaired haematopoiesis [44], and aberrant STAT3, PI3K, Akt and NF-κB signalling leads to megakaryocyte secretion of fibrotic cytokines including IL-1β, fibronectin, vascular endothelial growth factor (VEGF), platelet derived growth factor (PDGF) and tumour growth factor (TGFβ) [45,46].

The vast majority of *JAK2* mutant MPNs is due to the *JAK2^V617F^* mutation; however. in rare cases of erythrocytosis mutations in *JAK2*, exon 12 have been identified affecting residues between K537 and E543. Mutations in exon 12 show high levels of phosphorylated JAK2 and STAT5 compared to wildtype and higher than *JAK2^V617F^* in the case of K539L substitution. The mutation confers cytokine-independent growth but, unlike *JAK2^V617F^,* is only present as heterozygous mutations. The mutation affects common myeloid progenitors but has significant erythroid skewing and is only present in peripheral blood granulocytes at low levels and hence can be easily missed when only peripheral blood is sequenced [47].

## 5. CALR (Calreticulin)

Mutations in the Calreticulin genes are found in 25% of ET and 36% of PMF [48,49]. Located on chromosome 19, at the p13.13 locus, the gene contains 9 exons and encodes for a multifunctional protein product with an N-terminal lectin domain, proline-rich domain and acidic carboxy (C-) terminal domain that terminates as a KDEL amino acid sequence [17]. Through its c-terminal domain, CALR acts as a major calcium binding and storage protein in the lumen of the endoplasmic reticulum and in the nucleus. The lectin chaperone component also assists folding of glycoprotein substrates to prevent protein aggregation [50]. CALR expression on the cell surface imprints a cell for phagocytosis by macrophages particularly through cleaving the KDEL sequence and binding to LRP1 for surface presentation [51]. The KDEL sequence acts as the key endoplasmic retention signal [48,49]. 

Mutations of *CALR* occur as either an insertion and/or deletion in exon 9 of the gene. This leads to a frameshift change, resulting in the loss of the normal KDEL sequence and thus a positively charged amino acid sequence at the C-terminus lacking an endoplasmic retention signal [48,49]. The positive charge of the C-terminus is required for CALR to bind to thrombopoietin receptor (MPL) and to induce a proliferative phenotype. Mutations that result in uncharged glycine residues or complete truncation of exon 9 removing the KDEL sequence and the entire C-terminus leads to total loss of transforming ability [52]. The CALR-MPL interaction is critical to transcriptional activation of STAT5 and can only occur at the cell surface as retention of CALR in the endoplasmic reticulum abrogates STAT5 activity [53,54]. The mutant CALR is able to form a homomultimer at the cell surface with MPL. This homomultimeric structure is required to activate MPL in the absence of a ligand as wildtype CALR lacks the ability to undergo these multimeric changes [55]. Despite the role of STAT5 in *JAK2*-driven MPN promoting a PV phenotype, *CALR* mutation is unable to induce polycythemia or ligand independence when co-expressed with the EPO receptor [52,56].

CALR mutations are divided into two types; the first are type 1 (52-bp insertion). and type 1-like mutations lead to complete loss of all negatively charged amino acids. The second is type 2 (5-bp insertion), and type 2-like mutations eliminate approximately half of the negatively charged amino acid sequences [57,58]. The type of mutation confers some phenotypic differences, with type 1 mutations seen more commonly in primary myelofibrosis and is associated with a lower DIPSS score, suggesting a more favourable prognosis in comparison to other forms of PMF. In contrast, type 2 mutations more commonly lead to ET and have higher platelet counts than type 1 *CALR* ET. However, in primary myelofibrosis, type 2 mutations confer a similar phenotype to *JAK2^V617F^*-positive MF and have more pronounced splenomegaly, circulating blasts and cytopenia than the type 1 *CALR* counterpart [57]. *CALR* is not associated with a polycythemia phenotype [49].

Unlike *JAK2, CALR* is a rare mutation in asymptomatic clonal haematopoiesis at 0.16% but comparatively higher variant allele frequency (VAF) at 7.5% (vs 2.1% in JAK2). The average platelet count was also higher in *CALR* CH compared to normal controls. In contrast to *JAK2*, the proportion of *CALR* CH is relatively small compared to the proportion of *CALR*-mutated MPN [29]. CALR mutation is associated with a rapid development of clonal dominance in comparison to *JAK2^V617F^,* with high allele burdens in granulocytes of patients in essential thrombocythemia [31]. High variant allele burden is common in *CALR* MPN with clonal dominance apparent throughout stem cell, progenitor and granulocyte compartments in both early MPN (essential thrombocythemia) and late stage myelofibrosis. This is in contrast to *JAK2* where the variant allele burden in the stem cell compartment is relatively small in early stage MPN (ET or PV) and much higher in primary myelofibrosis and late stage disease (post PV/post ET MF) [59].

Recently, Nam et al. showed that the effect of *CALR* increases with myeloid differentiation, with higher VAF seen in HSPCs compared to progenitors. Within ET, MkPs show increased cell cycle gene expression and upregulated unfolded protein response (UPR) genes such as *IRE1* compared to haematopoietic stem cells (HSCs). In comparison, MF shows no difference in gene enrichment between MkPs and HSCs. Additionally, mutant MkPs show upregulated TGFβ with the degree of upregulation correlating with the degree of fibrosis seen in MF and ET [60]. Mutant *CALR* is found in 9–20% of leukemic transformation from MPN [61,62,63] and 25% of post-ET MF consistent with the prevalence of *CALR*-mutated ET [64].

## 6. MPL (Myeloproliferative Leukaemia)

Mutations in *MPL*, located on chromosome 1 p34, account for 1–3% of ET and 5% of MF [65]. The gene contains 12 exons with 2 cytokine receptor domains, transmembrane domain and a cytoplasmic domain. It is named after the murine retrovirus, myeloproliferative leukaemia virus (MPLV) [18]. The *MPL* gene encodes for thrombopoietin receptor protein (TPOR). The normal ligand of TPOR is thrombopoietin with key receptors on HSC, MPP, common myeloid progenitors (CMP), megakaryocyte-erythroid progenitors (MEP) and MkP in additional to receptors on megakaryocytes and platelets [66]. TPO binds to MPL/TPOR and induces receptor homodimerisation that subsequently leads to STAT5 phosphorylation and MAPK signalling [67]. MPL and TPO are critical to HSC self-renewal and expansion in addition to the proliferative effects TPO is able to stimulate DNA repair via the MPL receptor. In mouse models using γ-irradiation, de Laval et al. were able to demonstrate the role of TPO in increasing DNA-PK-dependent non-homologous end-joining (NHEJ) efficiency ensuring chromosomal integrity [68]. Whilst exogenous TPO agonists and endogenous TPO can promote DNA repair [69], the double strand break (DSB) repair function is dependent on an intact MPL receptor at both alleles with *Mpl*-deficient and haploinsufficient mice showing decreased NHEJ activity [68]. The above findings have been replicated by Barbieri et al. and further demonstrated the ability of MPL activation to promote an interferon-like antiviral gene response in HSCs [70].

There are several mutations of *MPL* identified, but the two main types occur within exon 10. These are mutations of tryptophan W515 located on the boundary of the transmembrane and cytosolic domains of MPL and lead to a leucine substitution W515L or a lysine substitution W515K [71]. Other mutations are much less common and include W515R, W515A and W515G [72]. Mutations are usually heterozygous, but homozygosity is seen with high mutations burdens and disease progression through copy neutral loss of heterozygosity of chromosome 1p. Variant allele burdens of greater than 50% are commonly associated with fibrosis both PMF and post ET-MF [73]. *MPL* mutations are mutually exclusive from *JAK2* mutations [65] and *JAK2^V617F^* shows antagonistic effects by downregulating MPL through enhanced ubiquitination of MPL and increased proteasome degradation [74]. *MPL^W515L^* mutations have also shown an upregulation of NHEJ pathways consistent with the effects of MPL receptor activation on DSB repair highlighting the ability of the MPN HSC clone to persist despite DNA damage [75].

*MPL* is critical at multiple stages of megakaryocyte development evidenced by knockout mice showing marked megakaryocyte deficiency and severe thrombocytosis. Haematopoietic progenitors were reduced and lack megakaryocytic potential [76,77]. 

MPL expression also acts as a regulator of TPO levels with mature platelets providing a negative feedback mechanism by removing TPO bound to MPL receptor (TPOR) [78]. Through altered MPL expression in late megakaryocytes and platelets, TPO is not cleared as readily leading to increased levels driving proliferation of early megakaryocytes. This leads to subsequent thrombocytosis as defective platelets are not able to provide the normal negative feedback loop [79].

## 7. Triple Negative MPN

Previously, a histological diagnosis of an MPN lacking a driver mutation was termed triple negative disease. Advances in sequencing technology have been able to identify novel mutations in *JAK2* and *MPL* reclassifying previously “triple negative” disease. Whole exome sequencing has not been able to identify a causative recurrent mutation within this group, and triple negative MPN likely represents a heterogenous group of clonal haematopoietic proliferation [80,81]. For this reason, triple negative MPN will not be explored further in this review.

## 8. Coexisting Mutations

The prognostic scoring systems for primary myelofibrosis have evolved from phenotype driven (IPSS/DIPSS) to now include information about additional gene mutations (DIPSS-plus and MIPSS, identifying high risk molecular lesions identified as mutations in *ASXL1, SRSF2, U2AF1, EZH2, IDH1* and *IDH2*) [82]. Whilst there has been some discrepancy in broad applicability across all MPNs [14], these highlight the prognostic significance of additional mutational burden. Mutations have been grouped according to the main functional category, being chromatin modifiers, spliceosome complex components, DNA methylation modifiers, tumour suppressors and transcriptional regulators (Figure 2). These mutations are described as coexisting mutations as their presence within a clone can precede the acquisition of the driver mutation or can occur subsequent to the driver mutation. We present a figure that shows the effect of additional mutation acquisition on disease progression, and represented in Figure 3, we propose that specific coexisting mutations drive progression to myelofibrosis and that other coexisting mutations favour leukemic transformation. Frequencies of each mutation in PV, ET, PMF, secondary MF (sMF) and leukemic transformation (LT) are included in Table 4.

## 9. Chromatin Modifiers

### 9.1. ASXL1 (Additional Sex Comb Like-1)

*ASXL1* mutations have significant impacts on prognosis and have been identified in 47% of MPN in the leukemic phase [61]. Mutations usually occur at the 5′ end of exon 12, mostly as frameshift mutations or less commonly nonsense mutations. Mutation leads to a truncated c-terminal of the resulting ASXL1 protein; this truncated form leads to a gain of function in some cellular pathways and loss of function in others [83]. Mutations are usually present as heterozygous in myeloid malignancy [84].

Normal ASXL1 has both silencing and enhancing effects on homeotic gene expression. It enhances methylation of histone marks through the polycomb repressor complex 2 (PRC2) that leads to repression and silencing of chromatin remodelling genes such as *HOXA* [85]. Through complexing with BRCA1-associated protein 1 (BAP1), ASXL1 forms the polycomb group repressive deubiquitinase complex, which is involved in stem cell pluripotency [86].

*ASXL1* mutation loss of function effects are mediated via a loss of PRC2-mediated histone H3 lysine 27 (H3K27) trimethylation, leading to loss of repression of the *HOXA* cluster of genes and loss of heterochromatin maintenance [85], whilst gain of function occurs through enhancing the activity of the ASXL1-BAP deubiquitinase complex, resulting in loss of histone H2K119 ubiquitination [87]. Together with loss of H3K27 trimethylation, there is upregulation of genes governing myeloid differentiation with significant myeloid skewing of LT-HSC [88]. This myeloid skewing can partially be attributed to binding of mutant ASXL1 to BRD4, resulting in the phosphorylation of RNA polymerase II, acetylation of H3K27 and H3K122 leading to upregulation of myeloid genes, and an increased sensitivity to BET bromodomain inhibitors [83,89]. Furthermore, mutant ASXL1 truncating protein expression results in more open chromatin in HSC (c-Kit+ cells) and dysregulated expression of genes critical for HSC self-renewal and differentiation [83].

*ASXL1* mutations are not limited to a specific molecular driver in MPN, with studies by Lasho et al. and McNamara et al. showing that, in accelerated and blast phase MPN, *ASXL1* mutations are present in *JAK2, CALR, MPL* and triple negative disease [61,62]. *ASXL1* mutations are more commonly enriched in leukemic transformation (LT) occurring from PMF, at a frequency of 66.7% compared with 13.6% of LT developing from PV, ET or sMF [63].

### 9.2. EZH2 (Enhancer of Zeste 2)

*EZH2* encodes a histone methyltransferase that constitutes the catalytic component of PRC2 functions to initiate epigenetic silencing of genes involved in cell fate decision. *EZH2* activity is particularly enriched with a trimethylation mark at nucleosomal histone H3K27 [90].

The majority of *EZH2* mutations in myeloid disease, particularly MPN, are missense loss of function and lead to premature chain termination or direct abrogation of histone methyltransferase domain [91]. This is a direct contrast to lymphoid malignancy, where *EZH2* mutations are highly expressed and lead to diffuse large B-cell lymphoma transformation from germinal centre B-cells [92,93]. Ernst at al. identified that deletion of 7q or loss of chromosome 7 leads to loss of the EZH2 tumour suppressor function in myeloid malignancy. The presence of monoalleleic mutations or uniparental disomy leading to homozygous mutations are responsible for oncogenic effects [90].

EZH2 expression is preserved in HSCs through MEPs but decreased in committed megakaryocytes. EZH2 loss in function produces megakaryocyte lineage features and high platelet counts in myeloid malignancy [94]. Commitment of HSCs or MPPs to specific lineages requires the removal of the H3K27me3 mark at the loci of master transcription factor regulators of lineage [95,96].

In combination with *JAK2^V617F^* mutations, mouse models with a loss of Ezh2 create an epigenetic switch in PRC target genes including *Hmga2, S100a8* and *S100a9* that increase colony forming units (CFU-Mk) committed to megakaryopoiesis [12,97,98].

*EZH2* mutations have been identified in 6–13% of myelofibrosis and are associated with poor survival [99]. However, its presence is less common in ET at 3% and absent in PV [7]. The presence of *EZH2* mutations is indicative of more progressive disease as highlighted above in myelofibrosis and by Lasho and McNamara in leukemic transformation with 15% and 7% identified in each cohort respectively [61,62]. *EZH2* mutation is also not specific to any particular driver mutation being identified in *JAK2, CALR, MPL* and triple negative MPN [16,61]. Whilst *EZH2* mutation were not identified in PV amongst the data sets reviewed, post-PV MF has demonstrated *EZH2* mutations signifying a phenotypic switch in PV inducing megakaryocyte proliferation and fibrosis consistent with *Ezh2* loss of function mouse models [64,97,98].

## 10. Spliceosome Complex Components

### 10.1. SRSF2 (Splicing Factor, Serine/Arginine-Rich, 2)

SRSF2 is critical for spliceosome assembly; it contains the ribonucleoprotein (RNP) type RNA binding motif and carboxyl-terminal serine/arginine-rich (SR) domain and binds to the exonic splice enhancer (ESE) with target pre-mRNA species [100].

Mutations in *SRSF2* involve a missense at proline 95 (P95H), with altered function leading to preferential recognition of CCNG ESE motifs compared to the wildtype product that recognizes both CCNG and GGNG ESE motifs equally. This alters the balance of splicing of numerous pre-mRNAs, leading to the production of miss-spliced products and nonsense-mediated decay [101,102]. SRSF2 P95H expression in mouse models leads to myeloid bias, impaired B cell development and maturation, and decreased haemopoietic stem cell numbers reminiscent of human myelodysplastic syndrome (MDS) in vivo [103]. MPN with *SRSF2* mutations have a high variant allele frequency, suggesting an impact on clonal dominance [104]. SRSF2 mutations are enriched in blast phase MPN (15–22%) and appear to be more common in combination with *JAK2* and *MPL* driver mutations compared to CALR [61,62]. There appears to be a particular predilection for *SRSF2* mutations in leukemic transformation after PMF at 50% compared to 6.8% seen in PV, ET and secondary MF-related AML [63].

### 10.2. U2AF1 (U2 Small Nuclear RNA Auxiliary Factor 1)

U2AF1 forms part of the spliceosome complex. The gene product is involved in pre-RNA splicing and recognizes pyrimidine-rich tracts with a conserved terminal AG present at 3′ splice sites [105]. Somatic mutations occur as heterozygous missense changes in serine 34 with an amino acid substitution to phenylalanine (S34F) or tyrosine (S34Y) [106]. Normally, serine 34 is highly conserved within the zinc finger domain critical for RNA binding. Mutant *U2AF1* leads to a splicing defect resulting in increased splicing, exon skipping and subsequent gain of function phenotypes [106]. 

Mutations in *U2AF1* represent less than 2% of chronic phase MPN [16], but their presence is associated with progression identified in 5–9% in leukemic transformation of MPN [61,62].

### 10.3. SF3B1 (Splicing Factor 3B, Subunit 1)

SF3B1 forms part of the spliceosome complex and has a prominent interaction with polycomb group proteins [107]. *SF3B1* mutations occur as missense mutations with the most common being K700E and H662Q amino acid substitutions [108]. Mutations of *SF3B1* block erythroid maturation [109], and whilst the mutation does not confer any additional prognostic significance in myelofibrosis [110], it commonly causes ring sideroblast changes to erythroblasts [111]. Consistent with the findings in MDS where *SF3B1* mutations generally confer a more benign phenotype, in MPN, it also carries a lower risk of leukemic transformation [108,111].

In combination with *CALR* mutations, *SF3B1* appears to increase the proliferative advantage of MkPs [60]. The subsequent aberrant splicing by SF3B1 leads to increased neoantigen formation. *CALR* mutants show an ability to bind common major histocompatibility (MHC) class I proteins, and the co-mutations with *SF3B1* showed the highest CALR neoantigen presentation on MHC I variants compared to other MPN mutations, suggesting a potential target for future immunotherapy [112]. 

## 11. DNA Methylation Modifiers

### 11.1. DNMT3A (DNA Methyltransferase 3A)

DNMT3A is part of the DNMT family of enzymes responsible for control of DNA methylation at CpG dinucleotides [113]. *DNMT3A* mutations are the most common mutation identified in clonal haematopoiesis [114,115]. 

The most common mutation of *DNMT3A* in myeloid malignancies is a missense mutation leading to substitution of arginine for histidine at position 882 (R882H). This compromises the enzymatic function, CpG specificity and flanking sequence preference of DNMT3A [116]. The degree of subsequent hypomethylation is relative to the mean CpG frequency per base of particular genes. Erythroid transcription factors such as *Klf1* and *Tal1* having higher CpG frequency per base show an increased erythroid transcriptional priming in *Dnmt3a* mutant cells in mice [117].

*DNMT3A* mutations are present in 5–10% of MPN and is present in early stage disease [67]. However, it is not represented in prognostic scoring systems [82]. Loss of *DNMT3A* function using CRISPR models in *JAK2^V617F^* positive models of MPN induces a myelofibrosis phenotype with reduced erythropoiesis, progressive depletion of HSCs and accumulation of MPPs. The transcription signature also shows aberrant self-renewal and inflammatory signalling pathways compared to *JAK2^V617F^* positive with wildtype *DNMT3A* [118].

### 11.2. TET2 (Ten-Eleven Translocation 2)

TET2 plays a key role in DNA methylation through oxidation of 5-methylcytosine (5mC) into 5-hydroxymethylcytosine (5hmC). This acts a stable epigenetic mark and is active in demethylation [113]. *TET2* contains 12 exons with mutations commonly occurring as nonsense or missense mutations leading to its loss of function [119]. Deletion or rearrangement of chromosome 4q24 is an additional mechanism for loss of TET2 function identified by Delhommeau et al. and further identified the mutation in early stage differentiation and its synergistic effects with *JAK2^V617F^* mutation [120].

Loss of normal TET2 function decreases 5hmC production, leading to DNA hypermethylation. This has a particular effect on HSCs increasing expression of self-renewal gene signatures, sensitizing haematopoietic cells to cooperating mutations and skewing towards myelomonocytic differentiation, [121] and sensitises haematopoietic cells to cooperation mutations. TET2 loss of function and subsequent hypermethylation patterns favour myelomonocytic lineage in HSC priming. Through increased expression of *Hlf, Sox4*, and *Meis1* there is increased self-renewal and increased susceptibility to hypermethylation of *Myc* and *Myb* due to their higher CpG-rich binding motifs [117]. *TET2* mutation can precede *JAK2* mutations in MPN HSCs and lead to biclonal disease or can occur secondary to *JAK2^V617F^* mutation and provide a competitive advantage driving clonal dominance [122]. *TET2* mutation order also influences disease biology in MPN, where *TET2* arising prior to *JAK2^V617F^* favoured an ET phenotype and *TET2* as a secondary mutation following *JAK2^V617F^* favoured a PV phenotype [123].

### 11.3. IDH1 and IDH2 (Isocitrate Dehydrogenase 1 and 2)

The isocitrate dehydrogenase enzyme genes are regulators of metabolic pathways that have indirect effects on epigenetic marks. IDH1 is a dimeric cytosolic NADP-dependent isocitrate dehydrogenase that catalyses decarboxylation of isocitrate into alpha ketoglutarate (α-KG) [124]. IDH2 is the mitochondrial version of the enzyme, produces NADPH and plays a key role in mitochondrial redox balance-mitigated oxidative damage [125]. *IDH* mutants increase production of 2-hydroxyglutarate (2-HG), which prevents histone demethylation. Preventing the expression of inducible lineage-specific differentiation genes leads to a block in differentiation that is characteristic of *IDH* mutant malignancies [126]. Mutations in *IDH1* occur as a missense mutation at codon 132, with the most common mutations being R132H and R132C, leading to an arginine substitution for histidine or a cysteine respectively [127]. Alterations to *IDH2* occur as missense mutations most commonly at codon 140 with R140Q the most commonly causing an amino acid substitution of glutamine replacing arginine. Mutations of codon 172 have also been described in transformed MPN with R172G [127,128].

The frequency of *IDH 1 and 2* mutations is low in chronic-phase MPN with numerous large cohorts identifying a frequency of between 0.5–4%, with the mutation seen more commonly in PMF than PV or ET [7,16,127,128,129]. There was no impact on overall survival for chronic phase MPN, but in contrast, the presence of *IDH* mutants in leukemic transformation conferred a worse overall survival than non-*IDH* mutated leukaemia [127]. The prevalence of *IDH* mutations is higher in leukemic transformation at 12.5–26% across several cohorts [61,62,63,127,128], suggestive of a strong risk for leukemic transformation with the acquisition of an *IDH* mutation.

McKenney et al. demonstrated the synergistic effect of *JAK2^V617F^* and IDH mutations, showing higher serum 2-HG levels for both *IDH1^R132H^* and *IDH2^R140Q^* models in combination with *JAK2^V617F^* compared to *JAK2* or *IDH* mutant alone. The combination of mutations showed impaired differentiation and increased immature progenitors compared to more late stage differentiated progenitors. Treatment with JAK2 inhibition was not able to significantly reverse the phenotype, but through combined inhibition of JAK2 and IDH2, the double mutant cells reverted to gene clustering patterns closer to WT LSK [130]. The presence of *IDH2* mutations has also been shown to enhance aberrant splicing of mutant *SRSF2*, leading to genomic instability and risk of leukemic transformation, highlighting yet another crosstalk between mutations [131].

## 12. Tumour Suppressors

### TP53 (Tumour Protein p53)

The tumour suppressor gene *TP53* is located on chromosome 17, encoding a DNA binding protein that responds to DNA damage by inducing transcriptional programs that result in cell cycle arrest or cell death via apoptotic pathways [132]. The presence of *TP53* mutations at low allelic frequency in chronic MPN with subsequent loss of heterozygosity is strongly associated with leukemic transformation [133]. This was confirmed by recent studies showing that *TP53* is more significantly enriched in leukemic transformation, found in 16–17% of cases compared to approximately 2% of chronic phase MPNs [16,61,62,134].

There are numerous upstream regulators of TP53; however, key in MPN are MDM2 and MDM4, which inhibit TP53 transcriptional function by facilitating nuclear export and by inducing ubiquitin-mediated degradation. MDM2 overexpression has been reported in MPN, and MDM4 high copy numbers are found in leukemic transformation and are the consequence of amplification of chromosome 1q [135,136].

*Trp53* loss leads to leukemic transformation following a short PV phase in *Jak2^V617F^* mouse models. Primitive leukaemia cell, MEP and CD71+ erythroid progenitors all retained the ability to cause leukaemia on serial transplantation including CD71+ cells, giving rise to erythroleukaemia. The progenitor population showed enhanced proliferative potential and resistance to apoptosis. Genomic instability was also increased with higher numbers of mice showing an abnormal karyotype compared to *Trp53* intact *Jak2* mutant mice [137,138]. *TP53* mutation appears to be a more common pathway to leukemic transformation in PV and ET compared to AML derived from PMF and secondary MF with 51.6% showing *TP53* mutations compared to 16% [63].

## 13. Transcriptional Regulators

### RUNX1 (Runt-Related Transcription Factor 1)

Located on chromosome 21q22.12, the *RUNX1* gene plays a critical role in haematopoiesis. All RUNX products contain a Runt homology domain (RHD) which is responsible for DNA binding and interaction with its common heterodimeric partner, CBFbeta. Through this DNA interaction RUNX1 controls the expression of target genes involved in haematopoietic differentiation, cell cycle regulation, ribosome biogenesis, and p53 and TGFβ pathways [139]. *RUNX1* mutations occurring in MPN are commonly heterozygous and occur as either missense, nonsense or frameshift mutations. The acquisition of *RUNX1* mutation is associated with leukemic transformation, with only 2.2% identified in chronic phase compared to 27.9% in transformed disease. *RUNX1*-mutated MPN has an adverse prognosis with median survival of 2.8 years compared to not reached (NR) for *RUNX1* wildtype [140]. The frequency of somatic *RUNX1* mutations is also higher in MPN-related AML compared to de novo AML with 17% vs. 5% in the cohorts examined by Lasho et al. [61].

## 14. Treatment of MPN—Existing and Emerging

### 14.1. Chemotherapy

Chronic long-term dosing with oral chemotherapy agents remain a mainstay of MPN therapy. The most widely used chemotherapeutic is the antimetabolite hydroxyurea (hydroxycarbamide). Hydroxyurea reduces deoxyribonucleotide production via inhibition of the enzyme ribonucleotide reductase through scavenging tyrosyl free radicals [141]. Used as a cytoreductive agent, hydroxyurea has been shown to decrease thrombosis rate in both PV [142] and ET [143]. In addition, reduction in splenomegaly and symptoms of MF have been observed although not sustained long term [144]. The limitations in the use of hydroxyurea has concerns regarding risk of leukemic transformation [145] despite some studies showing that the risk is not significantly higher than the background risk for MPN [146,147]. Often used as a comparator arm for other therapies, hydroxyurea can rarely show haematological and molecular responses [148]. Other oral agents include busulfan and pipobroman. Busulfan, an alkylating agent, has shown 80% haematological and 30% molecular response in hydroxyurea-resistant PV patients [149] but has conflicting data about whether it significantly increases risk of leukemic transformation [145,150]; hence, it is often used second line for shorter durations. Whilst European Leukaemia NET (ELN) has a definition for resistance to hydroxyurea, this does not incorporate molecular factors. Recently, p53-dependent mechanisms were identified as a resistance mechanism to hydroxyurea in colon cancer cells, highlighting the importance of DNA damage pathways in hydroxyurea sensitivity [151].

### 14.2. JAK Inhibitors

Inhibition of Janus kinase enzymes has emerged as the main target in drug development in MPN. JAK inhibition to date has shown clinical benefit independent of *JAK2* mutations [152,153] through potent inhibition of JAK1 and JAK2. JAK1-dependent cellular pathways are also activated in myelofibrosis and the downregulation of TNFα, IL-6, and MIP-1β through JAK1 and JAK2 blockade and STAT3 primarily through JAK2 blockade leads to the clinical benefits of decreased symptoms and splenomegaly [154]. JAK inhibitors exert their effect by either type 1 or type 2 inhibition of JAK2. Type 1 inhibition targets the active ATP binding pocket in a competitive manner, stabilizes JAK2 and increases activation loop phosphorylation [155]. Tvogerov et al. recently showed that type 1 JAK inhibition leads to no detectable STAT3 or STAT5 phosphorylation and minimal STAT1 phosphorylation. However, type 1 JAK inhibition prevented dephosphorylation and ubiquitination of JAK2, leading to progressive accumulation of an activation loop phosphorylated pool. The phosphorylated pool is rapidly depleted on JAK inhibitor cessation, accounting for the rapid recrudescence of the clinical syndrome known as JAK inhibitor withdrawal syndrome [156].

Type 2 JAK2 inhibition occurs when the inhibitor binds to the inactive state at a hydrophobic pocket adjacent to the ATP binding site uncovered by conformational change of the activation loop leading to stabilization of inactive JAK2 [155,157,158]. There is an absence of phosphorylation accumulation and hence no washout seen with type 2 inhibitor withdrawal in vitro [156]. JAK2 inhibitors currently in clinical practice and clinical trials include type 1 inhibitors ruxolitinib, fedratinib, momelotinib and pacritinib [158,159]. Type 2 JAK inhibitors BBT594 and CHZ868 has been limited to in vitro and PDX models with no published clinical trial data to date [160,161]. Chronic exposure to JAK2 inhibitors stabilizes activated JAK2 and increased JAK2 mRNA expression facilitates formation of a heterodimer of activated JAK2, JAK1 or TYK2 leading to JAK/STAT signalling escape. Removal of the JAK inhibitor may lead to resensitization by decreasing the drive to heterodimerisation [157].

### 14.3. Ruxolitinib

Ruxolitinib emerged as the first JAK targeted therapy for myelofibrosis following the results of the COMFORT-I and COMFORT-II phase 3 clinical trials. COMFORT-I showed a ≥50% improvement in symptom score and 41.9% of patients showed a ≥35% spleen volume reduction (SVR) at 24 weeks, whilst COMFORT-II showed that 28% of patients with a ≥35% (SVR) at 48 weeks had a mean reduction in spleen volume of 56% [152,153]. Subsequent analysis showed long-term survival benefit for ruxolitinib with a median overall survival of 5.3 vs. 2.4 years for best available therapy (BAT) after censoring for crossover [162].

The use of ruxolitinib has expanded to PV and ET, particularly to control disease in patient resistant or intolerant of hydroxyurea. In PV, ruxolitinib has proven to be superior to BAT for haematocrit (Hct) control, spleen size reduction and complete haematological response (CHR) [163,164,165,166]. RESPONSE-1 showed Hct control at 60% for ruxolitinib compared to 20% in the BAT arm, an ≥35% SVR of 38% vs. 1% and a CHR of 24% vs. 9% [163]. The results in ET have not shown the same differences between ruxolitinib and BAT with CHR rates of 46.6% vs. 44.2% in MAJIC-ET and no significant difference in molecular responses at 12 months [167]. However, the reduction in allele burden for *JAK2^V617F^* was apparent in a separate study with ongoing therapy showing a median decrease of 2.8% at 24 weeks improving to a 60% reduction by week 312 [168].

Ruxolitinib does not reliably eradicate clones harbouring JAK2V617F or CALR mutations in myelofibrosis, with only 12% of patients having a >50% reduction in VAF of JAK2 in the COMFORT-I cohort [169]. Similarly, a significant change in VAF was not seen for CALR in the COMFORT-II cohort [170]. This is in contrast to findings in polycythemia vera with a slight reduction in mean *JAK2* allele burden reduction in RESPONSE for the ruxolitinib treatment arm, strongly correlating with spleen reduction of >35% [171]. Type 1 JAK2 inhibitors such as ruxolitinib or fedratinib are unable to eradicate the disease-initiating *JAK2^V617F^* mutant LT-HSCs [172,173]. This is due a preferential selection for more committed MPP and MEPs with an inability to significantly block STAT1 phosphorylation in LT-HSCs. Interestingly ex vivo treatment with ruxolitinib was able to completely block STAT1, suggesting an inadequate intracellular drug concentration in LT-HSCs [172]. 

Clinical data reveal that response to JAK2 inhibition is impaired with concurrent *ASXL1* and *EZH2* mutations (hazards ratio of 2.94) and that 3 or more high molecular risk mutations have a shorter time to treatment failure [174]. Resistance to JAK inhibitors has been a complication of long-term therapy with loss of spleen response and disease progression with 50% of patients discontinuing ruxolitinib at 3 years due to side effects or loss or response [175]. 

### 14.4. Fedratinib

The 2nd JAK inhibitor approved by the U.S Food and Drug Administration (FDA) for myelofibrosis [176], fedratinib, is a selective JAK2 inhibitor with no significant JAK1, JAK3 or TYK3 inhibition [177]. The degree of immunosuppression appears less due to its selectivity for JAK2 with no effect observed on the expression of inflammatory cytokines IL-1α, MCP-1, E-selectin and P-selectin or immunomodulators sIL-17f and HLA-DR [159]. Despite phase 2 and 3 studies opening in 2011–2012, fedratinib underwent a clinical hold from 2013–2017 due to concerns regarding Wernicke encephalopathy for which it now has a black box warning [177]. The principal studies showing efficacy were JAKARTA-I and JAKARTA-II. JAKARTA-I examined fedratinib in frontline therapy for myelofibrosis showing a 36% response in ≥35% SVR and 36% response for a ≥50% improvement in total symptom score compared to placebo [178]. JAKARTA-II was a single arm phase 2 study of fedratinib in patients intolerant or refractory to ruxolitinib; the results showed a 55% response for ≥35% SVR and 26% for improvement of ≥50% in total symptom score [179]. Whilst there is no mutational data reported regarding response to fedratinib, further analysis of JAKARTA-I and II and subsequent studies FREEDOM-I and II may further identify these [180].

### 14.5. Momelotinib

Momelotinib, a second generation JAK2 inhibitor, has shown efficacy in myelofibrosis in SIMPLIFY-I, a phase 3 non-inferiority study against ruxolitinib. At 24 weeks, 26.5% in the momelotinib arm compared to 29% in the ruxolitinib arm had achieved ≥35% SVR. Whilst showing the benefit of greater transfusion independence (66.5% vs. 49.3% at 24 weeks), momelotinib failed to achieve non-inferiority for symptom control (≥50% reduction in symptom score was 28.4% for momelotinib vs. 42.2% for ruxolitinib) [181]. Following this, SIMPLIFY-II was unable to demonstrate superiority over BAT, 89% of which was patients on ruxolitinib [182]. The difference in efficacy between momelotinib and ruxolitinib might may be due to difference in anti-inflammatory effects particularly a lack of IL-1α suppression from momelotinib [159]. This lower level of suppression over pro-inflammatory signals may account for the lower symptom score reductions seen in SIMPLIFY-I. SIMPLIFY-I and SIMPLIFY-II were both able to demonstrate higher levels of transfusion independence than the comparator arm, and this is due to the ability of momelotinib to bind to bone morphogenic protein receptor kinase, activating A receptor type 1 (ACVR1). This receptor can increase hepcidin expression from the liver, and its inhibition improves iron utilization for erythropoiesis [183]. Long-term follow up of momelotinib therapy in myelofibrosis has shown a median overall survival of 3.6 years and 1.5 years in the high-risk group. Mutations in *ASXL1* and *SRSF2* were among the predictors of poor survival post-momelotinib therapy [184].

### 14.6. Pacritinib

Pacritinib is a selective JAK2 inhibitor that also exhibits activity against FLT3. Despite negligible JAK1 activity pacritinib maintains IL-1β suppression through inhibition of IL-1 receptor-associated kinase 1 [185]. The lack of significant JAK1 activity and suppression of IL-1β abrogates some of the myelosuppressive effects of other JAK inhibitors. The PERSIST-1 trial using pacritinib compared to BAT (excluding prior and current JAK inhibitors use) in myelofibrosis showed an increased in platelet count and a greater percentage of transfusion-independent patients at 24 weeks compared to BAT. Pacritinib also showed that 19% of patients achieved ≥35% SVR and that 36% achieved a ≥50% reduction in total symptom score. Reduction in allele burden was also measured at 24 weeks with pacritinib achieving a 15.8% reduction compared to 7.9% in BAT [186]. PERSIST-2 compared pacritinib to BAT but allowed prior ruxolitinib exposure and ruxolitinib treatment in the BAT arm. Pacritinib showed efficacy in 18% of patients achieving ≥35% SVR compared to 3% of BAT and 25% achieving ≥50% reduction in total symptom score compared to 14% (although not statistically significant); 48% of patients had received prior ruxolitinib, and 45% within the BAT arm received ruxolitinib. The improvement in transfusion and thrombocytopenia was also observed with pacritinib in PERSIST-2 [187]. Despite these results, pacritinib is not yet approved by the FDA or the European Medicines Agency (EMA) due to toxicity concerns. Trials are currently ongoing determination of the extent of toxicity and exploring dose modifications due to concerns from early reports of intracranial haemorrhage and cardiac failure identified in the PERSIST studies [188]. 

### 14.7. Pegylated Interferon Alpha (IFNα) 

The use of interferon in MPNs extends back to 1988 with IFNα used as a treatment for ET [189]. The clinical effects seen occur via its effectors, interferon stimulated genes (ISG). The expression of ISG subsets link to interferon regulatory factor (IRF), NF-κB and other transcriptional regulators. Endogenous interferons are defined as type 1, type 2 and type 3, with type 2 IFNs particularly IFNγ involved in the inflammatory response inducing JAK1 and JAK2 phosphorylation to activate STAT1. Type 1 IFN (IFNα and IFNβ) uses JAK1 phosphorylation and Tyk2 to activate STAT1 and STAT2 and the induction of ISGs [190,191]. In mouse models, IFNα activates dormant HSC by increased phosphorylation of STAT1 and PKD/Akt with a subsequent upregulation of IFNalpha, ISGs and Sca1. Acute exposure promotes cell proliferation from a quiescent state, whereas chronic exposure impairs HSC function and leads to progressive depletion of cycling HSCs through provoking DNA damage [192,193]. HSCs can utilize interferon regulatory factor-2 (IRF2) to protect quiescent stem cells from type 1 IFN-dependent exhaustion. IRF2 is able suppress IFNα signalling to preserve self-renewal and differentiation capacity of HSCs [194]. 

Treatment with IFNα stimulates both *JAK2^V617F^*-positive and WT quiescent stem cells into cycle but leads to a preferential depletion of *JAK2^V617F^* LT-HSCs in mice [195] and inability to develop MPN with serial transplantation with *JAK2^V617F^* mutant cells subsequent to IFNα treatment [196]. We have previously demonstrated enhanced cell cycle activity at baseline and the increased sensitivity to IFNα-mediated cycle activation of *JAK2* mutant cells. The mechanism for this increased sensitivity is not limited to increased cell cycling with other mechanism possibly implicated [195]. Recently, this has been further clarified with *JAK2^V617F^* LT-HSC showing increased STAT1 phosphorylation, reactive oxygen species (ROS) production and DNA damage in response to chronic IFNα treatment. The effect is more marked in *JAK2* mutant LT-HSCs compared to wildtype [172]. 

Interferon has shown favourable response rates in PV, with 94.6% of patients achieving CR at 12 months; 89.6% of patients had a reduction in *JAK2^V617F^* VAF with 24% achieving a molecular CR with an undetectable JAK2 clone [197]. In a similar study, 70% of PV and 76% of ET patients achieved CRs with 54% and 38% showing a *JAK2* molecular response and 14% and 6% of PV and ET respectively showing undetectable *JAK2^V617F^* clone [198]. The molecular response with IFNα in PV improves over time with 66% achieving molecular response in *JAK2* at 36 months of roPEG-IFNα-2b compared to the decreasing molecular response seen in the control group of best available therapy [148]. 

Interferon is also active in myelofibrosis with a recent FIM observational study showing that 58% of patients reduced their *JAK2* allele burden by more than 50%. The median overall survival for the cohort was 7.4 years, with a strong correlation between overall survival and longer duration of IFNα therapy [199]. In comparison to pooled data on overall survival from COMFORT-I and COMFORT-II, the FIM study showed overall survival for high risk and INT-II DIPSS risk as 4.6 and 6.9 years, respectively, whereas the pooled data showed an overall survival of 4.2 years for high risk and estimated OS of 8.5 years [162,199]. 

Concomitant *TET2* or *DNMT3A* mutations are associated with molecular resistance to interferon [200]. The FIM study also showed inferior response to IFNα in patients with high molecular risk (HMR) mutations (defined as *ASXL1, SRSF2, EZH2* and *IDH1/2*) [199]. Response to IFNα is not only restricted to *JAK2* mutant MPN, with *CALR* mutated ET also showing a favourable response to IFNα. IFNα leads to a haematological response in all patients and a median decrease in allele burden from 41% to 26%, with 42% of patients showing a greater than 50% reduction in *CALR* allele burden. The comparator arm of hydroxyurea or aspirin only did not show any significant reduction in allele burden. A suboptimal response was seen in 6 patients and was attributed to the presence of additional mutations of *TET2, ASXL1, IDH2* or *TP53* [201].

Clinical trials testing the combination of IFNα and ruxolitinib provide an interesting conundrum. On one hand, interferon does increase phosphorylated STAT1 to induce LT-HSCs out of quiescence. Ruxolitinib may be predicted to block STAT1 activation through inhibition of JAK1. We have recently shown that this potential antagonism is avoided in vivo because of the relative sparing of LT-HSC from ruxolitinib-mediated JAK/STAT inhibition [172]. Initial combination phase 2 studies showed a complete haematological response was achieved in 44% of PV patients and 58% of low to intermediate-1 risk MF at 12 months of treatment, with both groups showing a significant reduction in *JAK2^V617F^* allele burden with a median decrease in VAF from 47% to 23.5% for PV at 12 months and 45% to 18% for MF at 12 months [202].

### 14.8. Allogeneic Bone Marrow Transplantation

The only definitive curative option for MPN treatment is allogeneic bone marrow transplantation which is used in both PMF and secondary MF. Transplant selection involves multiple disease-related, recipient-related, donor-related and institutional factors that extend beyond the scope of this review. However, in the setting of myelofibrosis, allogeneic transplant is limited to patients less than 70 years of age and with an estimated survival less than 5 years [203]; this incorporates DIPSS classification of intermediate-2 and high risk based on comparisons between transplant and no transplant survival in the ruxolitinib era. Transplant in intermediate-1 disease could be considered if *ASXL1* mutation or >2% blasts are present [204]. The emphasis of molecular risk in MIPSS+ 2.0 and GIPSS and the secondary MF-specific MYSEC-PM are acknowledged as enhanced tools to guide transplant decision in this patient group [203].

A recently published analysis of 1055 patients transplanted for MF showed an overall survival of 74% at 10 years with a disease-free survival at 2 years of 64%. This was compromised by excess mortality of 33% in patients ≥65 years and 14% in patients <45 years, identifying male gender, age, secondary myelofibrosis and lack of any graft versus host disease (GVHD) as risks for relapse and excess of mortality [205]. Comparison of haematopoietic cell transplant (HCT) vs. no HCT in MF was recently evaluated, showing a long-term benefit for allograft but a significant risk of transplant related mortality (TRM) within the first year. Overall survival at 1 year favoured non-HCT with a hazard ratio (HR) of 0.26 for Int-1 and 0.39 for Int-2 and high risk. However, after this initial 1-year period, HCT showed superior outcomes with an HR of 2.64 for Int-1 and 2.55 for Int-2 and high risk [206]. Interestingly, most prognostic mutations such as *ASXL1, TP53, SRSF2, IDH1/2* and *EZH2* do not alter overall survival or relapse free survival following allogeneic transplant, with only *U2AF1* or *DNMT3A* having adverse effects on overall survival [207].

For further guidance regarding allogeneic transplant in myelofibrosis, the reviews by Devlin and Gupta [208] and the EBMT Handbook [203] are recommended. 

Figure 4 provides an overview of treatment modalities and proposed mechanism of these treatments in MPN.

## 15. Future Directions 

### 15.1. Checkpoint Inhibitors and Cellular Therapies

*JAK2* and programmed death ligand 1 (*PD-L1*) genes are co-located on chromosome 9p24, and recently, Prestipino et al. demonstrated that PD-L1 expression is increased in T cells, megakaryocytes and monocytes of mice with *JAK2^V617F^* mutations compared to wildtype controls. They went on to show the ability of STAT3 and STAT5 to bind to the Cd274 promoter encoding PD-L1 and specifically that STAT3 is the main effector mechanism to upregulate Cd274 expression. PD-L1 expression could be subsequently reduced by JAK or STAT3 inhibition. Anti-PD1 antibody therapy was able to successfully reduce the *JAK2^V617F^* allele burden in transplanted mice but showed that this was strongly dependent on an intact donor T cell response [209]. Further efficacy of PD-L1-based therapy in human trials remains to be seen, but this work shows early proof of concept.

Additionally, T cell responses against mutant JAK2 have been identified. Through identification of HLA-A2-restricted epitopes in the *JAK2^V617F^* mutation, Holmström et al. were able to induce a specific CD8+ cytotoxic T-lymphocyte culture using a JAK201 peptide loaded onto autologous dendritic cells [210]. A similar study with *CALR* mutant clones has generated HLA-DR-restricted specific CD4+ T-lymphocytes able to target mutated *CALR* [211], further highlighting a growing potential for cellular based therapies to develop for MPN.

### 15.2. Ongoing Clinical Trials

Currently, there are several phase 1–3 clinical trials ongoing in MPN that extend beyond the scope of this review, but they include histone deacetylase (HDAC) inhibitors, modulators of cellular apoptosis such as navitoclax and BH3-mimetics, monoclonal antibodies such as tagraxofusp, hypomethylating agents, and combination therapies such as ruxolitinib and lenalidomide [12,212,213].

## 16. Summary and Conclusions

Myeloproliferative neoplasms have benefitted from significant research into basic biology providing insight into the pathogenesis of these diseases. Whilst there are treatments that control the disease, these is still an absence of a “game changer” that can abrogate the adverse risk features or eradicate the disease in the majority of patients. Leukemic transformation continues to remain a feared complication of MPN, and whilst we can identify the risks for transformation, we still lack any way of stopping it outside of an allograft that is limited to a specific group of patients. Further research into novel combinations, agents that can alter the natural history of disease, decreasing the risk of MF progression or leukemic transformation and overcoming adverse genetic risk features are desperately needed to improve the survival of patients with MPN.

## Figures and Tables

**Figure 1 cells-09-01901-f001:**
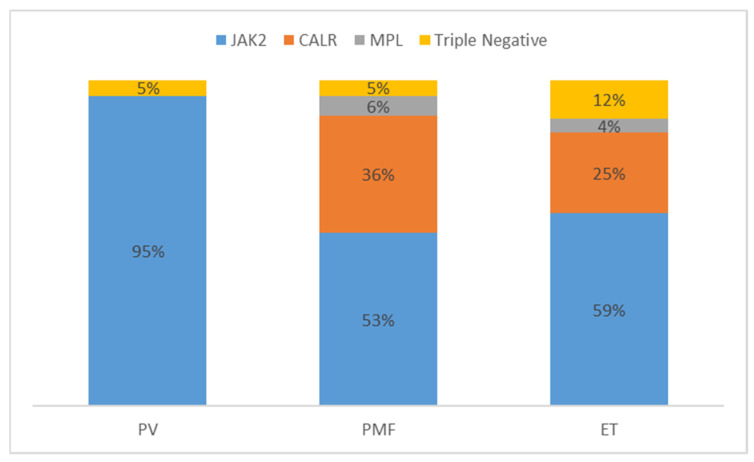
Adapted from [17] showing the frequency of each driver mutation in Myeloproliferative neoplasms (MPN) relative to the disease phenotype.

**Figure 2 cells-09-01901-f002:**
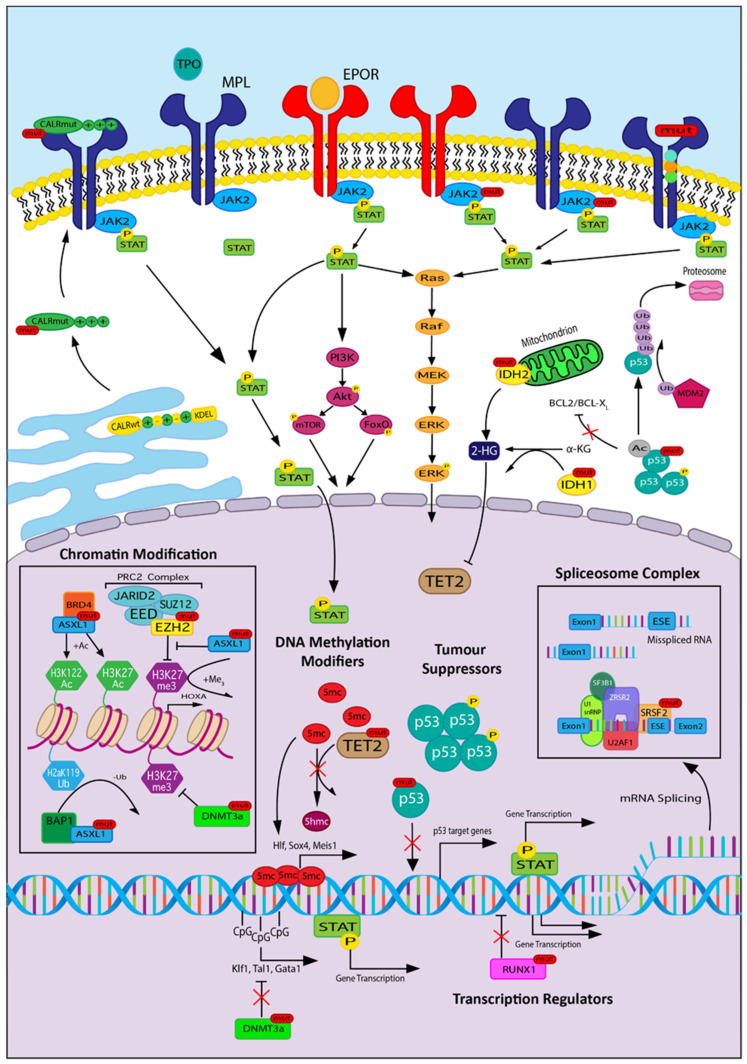
The molecular signalling pathways involved in MPN: The cell surface receptors are erythropoietin receptor (EPOR; red) and thrombopoietin receptor/MPL (navy blue), MPL with mutated Calreticulin (CALR), wildtype MPL with no ligand (thrombopoietin (TPO)) bound and no STAT signalling, wildtype EPOR with bound ligand (erythropoietin (EPO)) leading to STAT signalling, EPOR with JAK2 mutant, MPL with JAK2 mutant and mutated MPL. The cytoplasm shows STAT pathway signalling with activation of phosphatidylinositol 3-kinase (PI3K)/Akt and RAS pathways, and the nucleus (lilac background) shows the effects of driver and coexisting mutations on nuclear functions. Headings for DNA methylation modifiers, tumour suppressors, transcription regulators, spliceosome complex and chromatin modification identify the key sites of coexisting mutations. Abbreviations not mentioned in the body of the article: STAT, signal transducers and activators of transcription; PI3K, phosphatidylinositol 3-kinase; Akt, Protein kinase B; mTOR, mammalian target of rapamycin; SUZ12, suppressor of zeste 12 homolog; EEZ, embryonic ectoderm development; BAP1, BRCA1-associated protein 1; BRD4, bromodomain containing protein 4; BCL-2, B cell lymphoma 2; BCL-XL, B cell lymphoma extra large; Ac, acetylated; Me, methylated; Ub, ubiquitinated; P, phosphorylated; mut, mutated.

**Figure 3 cells-09-01901-f003:**
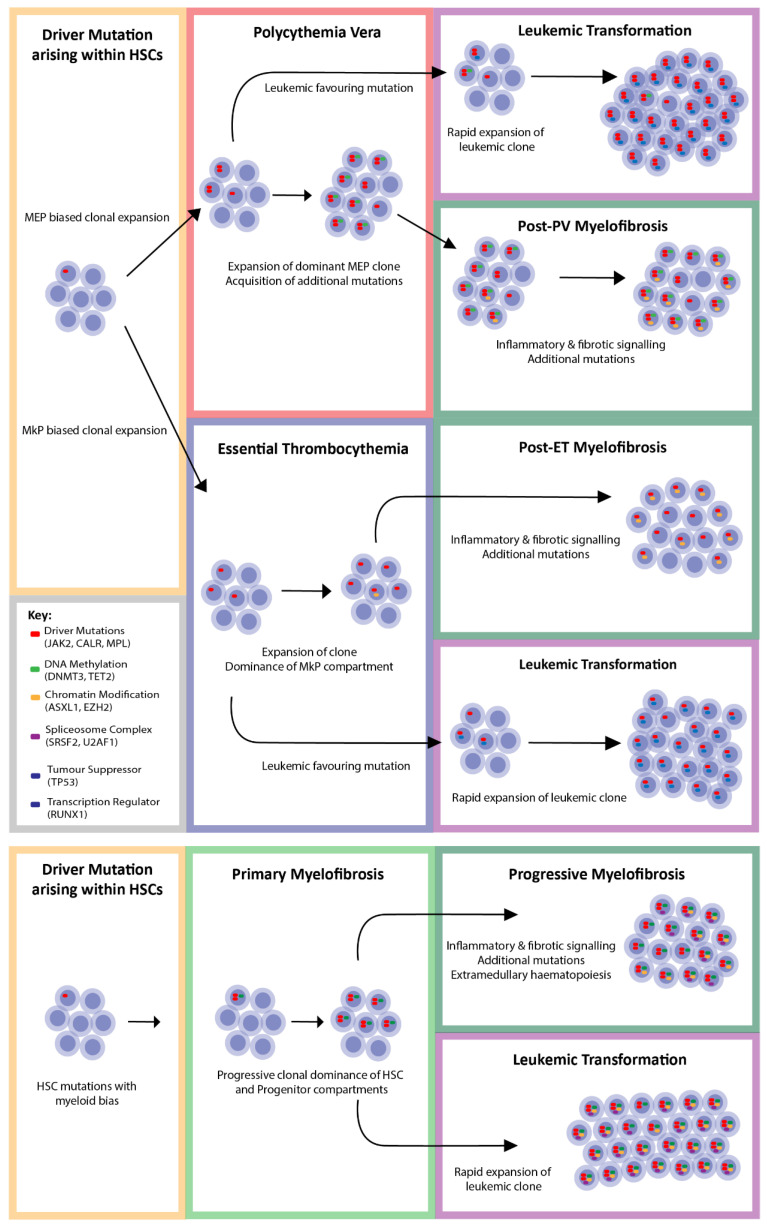
A proposed model for clonal evolution in MPN with acquisition of additional mutations leading to disease progression.

**Figure 4 cells-09-01901-f004:**
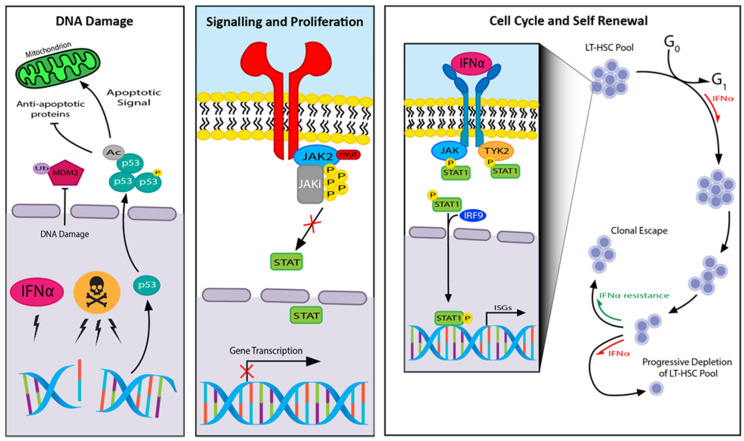
Treatments in MPN and the mechanism of action to control the disease through DNA damage, signalling and proliferation and cell cycle and self-renewal effects: Toxic symbol, chemotherapeutic agents; JAKi, JAK inhibitor; IRF9, interferon regulatory factor 9; ISGs, interferon-stimulated genes; TYK2, tyrosine kinase 2.

**Table 1 cells-09-01901-t001:** Adapted from [9] showing incorporation of molecular risk factors and median overall survival (OS) for respective risk groups; Int, Intermediate.

Scoring System	Criteria	Low Risk	Int Risk	High Risk
MIPSS-PV	Age > 67 years (2 points) Leukocyte count ≥15 × 10^9^/L (1 point) Adverse Mutation—SRSF2 (3 points) Thrombosis history (1 point)	0–1 point Median OS 24 years	2–3 points Median OS 13.1 years	≥4 points Median OS 3.2 years
MIPSS-ET	Age > 60 years (4 points) Male gender (1 point) Leukocyte count ≥11 × 10^9^/L (1 point) Adverse Mutations—SRSF2, SF3B1, U2AF1, TP53 (2 points)	0–1 point Median OS 34.4 years	2–5 points Median OS 14.1 years	≥6 points Median OS 7.9 years

**Table 2 cells-09-01901-t002:** Mutation enhanced international prognostic scoring system (MIPSS70-Plus) adapted from [12] which can be applied at any time point of the disease: very high risk karyotypes include −7, inv(3)/3q21, i(17q), 12p-/12p11.2, 11q-/11q23 or autosomal trisomies other than +8 or +9; unfavourable karyotypes include any karyotype other than normal or sole abnormalities of 20q-, 13q-, +9, chromosome 1 translocation/duplication, -Y or sex chromosome abnormality; HMR, high molecular risk mutations include *ASXL1, SRSF2, EZH2, IDH1, IDH2* and *U2AF1Q157*; NR, not reached; Int, intermediate.

Scoring System	Criteria	Very Low Risk	Low Risk	Intermediate Risk	High Risk	Very High Risk
MIPSS70-Plus Version 2.0 (Any time point)	Severe Anaemia (2 points) (Hb < 80g/L female, < 90g/L male) Moderate Anaemia (1 point) (Hb 80–100g/L female, 90–110g/L male) Circulating blasts ≥ 2% (1 point) Constituitional symptoms (2 points) Very high risk karyotype (4 points) Unfavourable karyotype (3 points) ≥2 HMR mutations (3 points) 1 HMR mutation (2 points) Type 1/like CALR absent (2 points)	0 points Median OS NR	1–2 points Median OS 16.4 years	3–4 points Median OS 7.7 years	5–8 points Median OS 4.1 years	9+ points Median OS 1.8 years

**Table 3 cells-09-01901-t003:** Dynamic International prognostic scoring system (DIPSS) adapted from [11], which is widely used for enrolment in clinical trials; Int, intermediate.

Scoring System	Criteria	Low Risk	Int-1 Risk	Int-2 Risk	High Risk
DIPSS (Any time point)	Age > 65 years (1 point) Constitutional Symptoms (1 point) Hb < 100g/L (2 points) Leukocyte count ≥25 × 10^9^/L (1 point) Circulating blasts > 1% (1 point)	0 points Median OS 14.6 years	1–2 points Median OS 7.4 years	3-4 points Median OS 4 years	5–6 points Median OS 2.3 years

**Table 4 cells-09-01901-t004:** Shown is the frequency of driver and coexisting mutations for polycythemia vera (PV), essential thrombocythemia (ET), primary myelofibrosis (PMF), sMF and leukemic transformation (LT). PV and ET are adapted from [7], PMF and sMF are adapted from [64], and LT is adapted from [61]. * Isocitrate Dehydrogenase 1 and 2 (IDH1 and IDH2) mutations were combined in the paper and have been reported as such. ^$^ Chromosomal abnormalities that deregulate TP53 function have also been described, including amplification of MDM2 on chromosome 1 and 17p deletion. Abbreviations: NA, non applicable.

Mutation Group	Gene	PV [7]	ET [7]	PMF [64]	sMF [64]	LT [61]
Driver Mutations	JAK2	98%	52%	62%	81%	60%
	CALR	0%	26%	22%	14%	21%
	MPL	0%	4%	5%	3%	13%
DNA Methylation	TET2	22%	16%	15%	39%	19%
	DNMT3A	2%	6%	9%	5%	3%
	IDH1	0%	0%	2% *	1% *	12%
	IDH2	2%	1%	NA	NA	7%
Chromatin Modification	ASXL1	12%	11%	48%	27%	47%
	EZH2	0%	3%	6%	14%	15%
Spliceosome Complex	SRSF2	3%	2%	14%	3%	13%
	U2AF1	0%	1%	17%	7%	5%
	SF3B1	3%	5%	13%	5%	7%
Tumour Suppressor	TP53 ^$^	1%	2%	6%	14%	16%
Transcription Regulator	RUNX1	2%	2%	3%	3%	17%

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
