# Peer review of "MPN: The Molecular Drivers of Disease Initiation, Progression and Transformation and their Effect on Treatment"

_cells, 2020, doi:10.3390/cells9081901_

Round 1
Reviewer 1 Report
The authors addresses every aspect of MPN from pathogenesis to in this comprehensive review of myeloproliferative neoplasms covering from pathogenesis to treatment including molecular target therapies. This helps readers with obtaining up to date knowledge about MPN.
Minor points:
- Gene symbols should be written in italic.
- Font style is different from L527 to L532.
- Font style is different from L641 to L643
- L641 this should be Figure 4 instead of Figure1.
Reviewer 2 Report
Grabek and al propose a comprehensive review about myeloproliferative neoplasms, going from the description of the prognostic scoring systems to the review about the functions of each driver, and to the description of their use as a target for treatment.
To my opinion, the field covered by this review is too large, so the data presented by the authors are a bit superficial in each part of the review, even if the whole document provides a good overview of the MPNs. I also regret that this review is a catalog of data, with very few novel ideas or concept proposed by the authors, so the target reader is more a young haematologist/scientist who search for a strating overview on NPMs than an expert scientist/clinician. I believe that each part of the manuscript could be a separate review (classification and prognosis, pathophysiology, treatment) with more conceptual insights.
Moreover, the NEJM from Grinfeld and colleagues, even if cited in the review, is not adequatly discussed in the first part of the review. This paper proposes a new classification of these diseases based mainly on the genomics data, and provides interesting data about the prognosis stratification in an individualized manner. it should be detailed in the first part of the review.
Other comments :
line 45 : what is PN ?
line 104 : each driver mutation (no "s")
figure 1 : the so called triple negative MPN should be more discussed. 2 papers published in Blood in 2016 (milosevic feenstra et al and cabagnols et al) proved that some of them are not clonal...
line 143 : what do the authors mean with the sentence starting by "PV homozygosity..." ?
About JAK2 : a short sentence about JAK2 mutations in endothelial cells, especially in splanchnic thrombosis and budd chiairi syndrome, could be appreciated
line 171 : maybe explicitely precis the role of the KDEL signal
line 177 : check the grammar (not sure that the punctuation is correct)
About MPL : some word about the role of MPL in the hematopoietic stem cells (doi: 10.1084/jem.20170997 or doi: 10.1016/j.stem.2012.10.012)
The concept of "secondary mutations" seems inappropriate, because it suggests that they arrive after the first ones, which is not always the case... the author should find another terminology
line 257 : check grammar "protein as has"
About EZH2 : maybe precise that the loss of function mutations are frequent in myeloid malignancies, but gain of function mutations are observed in lymphoid malignancies. the authors should also discuss haploinsufficiency of EZH2 mediated by del 7q.
About DNMT3A : the authors should precise that this is the most frequently altered gene in clonal hematopoiesis.
About TET2 : the authors should cite the appropriate reference from Delhommeau et al (NEJM)
Figure 3 proposes a model of disease evolution which is not really discussed in the text...
Figures 4 and 5 could be improved.
Author Response
See attached document.

Reviewer 3 Report
Grabek and colleagues submitted a comprehensive review on the molecular and clinical aspects of myeloproliferative neoplasms.
The paper is written well with many informative figures and tables, however, it need modifications before acceptance for publication.
Specific comments:
- It would seem more logical to start with the part on the driver mutations followed by the prognostic systems.
- The authors could consider inclusion of the MPN Personalised Risk Calculator (https://www.sanger.ac.uk/science/tools/progmod/progmod/)
- JAK2 exon 12 mutations would deserve a more detailed description. There is only one sentence in the manuscript in connection with these rare variants.
- Pipobroman therapy is not included in the most recent guidelines. The authors should reconsider inclusion of this therapeutic modality
- Momelotinib is spelled as momelitinib throughout the manuscript.
- In connection with Pacritinib, the authors should mention that the PERSIST trial did not lead to FDA or EMA approval due to toxicities with new ongoing trials in progress.
- Finally, the future directions parts should be expanded. For instance, there are promising results on immune-checkpoint inhibitors, in MPNs for instance. The ongoing clinical trials would perhaps deserve a table summarising these efforts.
Author Response
See attached document.

Round 2
Reviewer 2 Report
I warmly thank the authors for having consider my remarks in this revised version of the manuscript. To my opinion, the review is now ready for publication
Reviewer 3 Report
No further comments.